# The Relationship between Drug Consumption and Dating App Use: Results from an Italian Survey

**Luca Flesia** [1,*], **Valentina Fietta** [2], **Carlo Foresta** [1] **and Merylin Monaro** [2]

1   Unit of Andrology and Reproductive Medicine, Department of Medicine, University of Padova, 35128 Padova, Italy; carlo.foresta@unipd.it
2   Department of General Psychology, University of Padova, 35131 Padova, Italy; valefietta.vf@gmail.com (V.F.); merylin.monaro@unipd.it (M.M.)
*   Correspondence: luca.flesia@ordinepsicologiveneto.it

**Abstract:** To date, the literature regarding the relationship between drug consumption and dating app use is still very scant and inconclusive. The present study was thus aimed at investigating the association between drug consumption and dating app use in the general population. A total of 1278 Italian respondents completed an online *ad hoc* questionnaire assessing drug consumption (cannabis versus other illicit drugs), dating app use, the primary motive for installing dating apps, and demographics. Multiple logistic regression analyses were run to investigate the role of demographics and dating app use on drug consumption. Being single predicted cannabis use. Using dating apps accounted for higher odds of cannabis use; however, people who intensely used the apps were less likely to consume marijuana. Conversely, dating app use was not associated with the consumption of other drugs. This study suggests the presence of common underlying factors between dating app use and cannabis use, and it highlights the mediating role of the intensity of app use. Conversely, the study suggests that dating app use and the use of other drugs are quite independent behaviors.

**Keywords:** cannabis; marijuana; illicit drugs; geosocial networking apps; mobile dating applications; motives

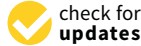

## 1. Introduction

During the past decade, the rise of mobile dating apps has revolutionized the way in which people construct social relations and find romantic and sexual partners. In fact, the number of dating app users has increasingly grown in recent years. According to the Statista Digital Market Outlook, the United States had 28.9 million users of online dating services in 2017, and this number increased to 44.2 million in 2020. It is estimated to reach 53.3 million by 2025 (Statista 2021). Globally, 270 million people worldwide used dating apps in 2020 (Curry 2021). Tinder, the most popular dating app, counts 1.6 billion "swipes" every day (Tinder 2019). Unlike traditional dating websites, which required lengthy profiles and complicated profile searches, geo-social dating apps are very easy to download and use. They enable users to select potential partners according to specific desired qualities (i.e., age, gender, weight, etc.), and they also allow users to find potential partners located in their spatial proximities. Finally, because these apps are installed on mobile devices, they can be used anywhere and anytime (Ranzini and Lutz 2017; Schreurs et al. 2020). Taking advantage on these affordances, dating apps provide users with a large potential of sexual partners instantly (Bickham et al. 2020). Accordingly, one of the first research questions addressed in the literature on dating apps concerned the impact of dating app use on users' sexual health, assuming an association between dating apps and sexual risk behaviors (Anzani et al. 2018; Wang et al. 2018; Ciocca et al. 2020; Flesia et al. 2021c). Some studies also investigated the association between dating app use and substance-related behaviors in conjunction with sexual experiences. Rogge et al. (2020) found that people who had used dating apps in the previous two months were more likely to have previously had

hookups involving recreational drugs compared with non-users of dating apps (Rogge et al. 2020). Choi et al. (2017) also found that using dating apps for more than one year was associated with drug use in conjunction with sexual experiences (both lifetime and the latest sexual experiences) (Choi et al. 2017).

With regard to the relation between drug consumption and dating app use outside of sexual experiences, the existing literature is still scant. Phillips et al. (2014), analyzing the frequency of drug consumption among a sample of "men who have sex with men" (MSM), found that app users reported a significantly higher use of crystal meth, poppers, and painkillers in the previous 12 months compared with non-users, whereas no differences were found regarding marijuana, cocaine, and heroin use (Phillips et al. 2014). Recently, Fansher and Eckinger (2020), investigating the association between Tinder use and risk behaviors in a sample of American university students under the age of 30, found that Tinder users were more likely to have consumed drugs during the past three months than Tinder non-users were (Fansher and Eckinger 2020). In the study, people who had used Tinder during the past were considered to be "app users". Erevik et al. (2020) also found that, among single Norwegian students, Tinder users were more likely to have used illicit drugs during the past six months compared with Tinder non-users (Erevik et al. 2020).

Although the literature seems to be consistent in indicating an association between dating app use and drug consumption, caution is needed in interpreting these results. Indeed, some studies considered "app users" to be only the people currently using the apps, whereas others considered "app users" to be both the people currently using the apps and the people who had used them in the past but no longer used them (other studies considered these to be "non-users"). Therefore, a distinction among the three subsamples of "active users," "former users", and "non-users" would provide more information. Moreover, most of the studies on this topic did not distinguish between the types of illicit drugs consumed. However, this is a significant distinction, especially between cannabis and other drugs. Indeed, cannabis use is much more common than the use of other drugs is, and although it is associated with health and psychosocial consequences, cannabis is generally seen as a "soft drug" within the general population due to the apparent low medical risks related to its use (WHO 2016; Verbanck 2018). Finally, none of the studies on this topic considered the association between drug consumption and dating app use in the general population. Rather, they investigated the role of demographic variables, such as sexual orientation, age, and relational status, in influencing this association. Therefore, a broader investigation of these links could expand the information and understanding on this specific topic.

According to the "Uses and Gratification" theory (Katz et al. 1973; Whiting and Williams 2013), individuals meet their personal needs when they use dating apps. The theory posits that various media, through their specific features, allow for different uses and gratifications. In alignment with this, the literature highlights that people can use dating apps for specific uses and to obtain specific gratifications (Griffin et al. 2018). Evidence shows that differences in motivations for installing the apps are associated with differences in behavioral patterns. For instance, while investigating the association between dating app use and smoking, Flesia et al. found that individuals using these apps with the primary aim of finding friends were less likely to smoke than other app users were (Flesia et al. 2021a). Similar results are reported with regard to the association between dating app use and alcohol consumption (Flesia et al. 2021b). Therefore, although substance use related to sexual intercourse is more readily linked to dating app use, a specific investigation into the association between dating app use and drug consumption, even outside of sexual activity, might provide useful information for implementing targeted prevention interventions, as well as pave the way for new lines of research.

The present study's first aim was to investigate the potential associations between dating app use and drug use in the general population. The study distinguished among active app users, former app users, and app non-users, and it also distinguished between cannabis and other drugs when it came to drug consumption.

Users' socio-demographic characteristics might influence the associations between dating app use and drug consumption. Young people are known to be more prone to use both dating apps (Sawyer et al. 2018) and drugs (Jackson et al. 2012) compared with the adult population. In addition, males, and even more young males, are known to be more prone to substance-related behaviors compared with their female counterparts (Becker and Hu 2008; Flesia et al. 2020). In the previously cited study by Fansher and Eckinger (2020), male users reported higher levels of recent illicit drug use compared with dating app non-users (Fansher and Eckinger 2020).

The present study's second aim was to investigate the role of some socio-demographics (i.e., age, sex, sexual orientation, relational status, educational level) in drug consumption.

Finally, the literature reports the role of patterns of dating app use (motives and intensity of use) in influencing their association with specific behavioral patterns (Flesia et al. 2021a). Among active app users, people who intensely used these apps were found to be less likely to smoke (Flesia et al. 2021a) and were more likely to engage in risky sexual behaviors (Rogge et al. 2020). The length of use might also influence the association between dating app use and behavioral patterns (Rogge et al. 2020). The present study's third aim was to investigate the influence of patterns of dating app use on the association between app usage and drug consumption.

**Hypothesis 1 (H1).** *Active app users, former users and non-users will differ in terms of their odds of cannabis use and the use of other illicit drugs. Active app users will show higher odds of drug consumption compared with former users and non-users.*

**Hypothesis 2 (H2).** *Socio-demographic variables will account for a portion of the variance in the relation between dating app use and both cannabis use and the use of other illicit drugs. Being male will account for a portion of the variance in drug consumption.*

**Hypothesis 3 (H3).** *The odds of drug use among active app users will differ according to their motives for dating app use. The odds of drug use among active app users will differ according to the intensity of use and length of dating app use.*

Considering the significant impact of substance use on global health in terms of the burden of disease and mortality (Peacock et al. 2018; Lim et al. 2012; Degenhardt et al. 2014), economic burden (Degenhardt et al. 2013, 2018; Gowing et al. 2015; Degenhardt et al. 2014), and the significant spread of dating apps (Pew Research Center 2016; Clement 2020; Statista 2020), an estimate of the possible interplay between dating app use and drug use might be very significant for programming targeted and effective drug prevention policies.

## 2. Materials and Methods

### 2.1. Participants and Procedure

Data were collected between 1 June 2019 and 30 September 2019. Participants were recruited cross-sectionally through an online link that was posted and advertised on social media. All participants were required to read and provide informed consent before beginning the online survey. Participation was voluntary. The study subjects completed a self-administered anonymous "ad hoc" questionnaire (see "Measures") managed through Google Forms. A total of 1390 respondents accessed the survey; 112 subjects were excluded, as 43 were aged less than 18 years, 40 did not complete the entire questionnaire, and 29 did not give informed consent. The final sample consisted of 1278 Italian-speaking participants.

The demographic features of the sample (number of participants, age, sex assigned at birth, educational level, sexual orientation, and relational status) are reported in Table 1.

**Table 1.** Demographic features of the total sample and the three split samples: non-users, former users, active users. (Percentages were rounded by excess from 0.05 up and by defect from 0.05 excluded down.)

| | Non-Users | Former Users | Active Users | Overall Sample |
|---|---|---|---|---|
| Number of subjects | 598 | 393 | 287 | 1278 |
| Age | Average = 26.35 (SD = 7.29) | Average = 27.70 (SD = 7.19) | Average = 31.60 (SD = 8.62) | Average = 27.94 (SD = 7.85) |
| Sex assigned at birth | Males = 127 (21.24%) Females = 471 (78.76%) | Males = 146 (37.15%) Females = 247 (62.85%) | Males = 191 (66.55%) Females = 96 (33.45%) | Males = 464 (36.31%) Females = 814 (63.69%) |
| Educational level (years) | Average = 15.02 (SD = 2.59) 8 years = 4.18% 13 years = 41.64% 16 years = 23.91% 18 years or more = 30.27% | Average = 15.19 (SD = 2.59) 8 years = 4.33% 13 years = 36.54% 16 years = 27.23% 18 years or more= 31.81% | Average = 15.57 (SD = 2.56) 8 years = 2.79% 13 years = 35.54% 16 years = 18.82% 18 years or more= 42.86% | Average = 15.20 (SD = 2.59) 8 years = 3.91% 13 years = 38.73% 16 years = 23.79% 18 years or more = 33.57% |
| Sexualorientation | Heterosexual = 85.12% Non-heterosexual = 14.88% Homosexual = 2.68% Other = 12.21% | Heterosexual = 56.49% Non-heterosexual = 43.51% Homosexual = 20.61% Other = 22.90% | Heterosexual = 34.15% Non-heterosexual = 65.89% Homosexual = 37.67% Other = 28.22% | Heterosexual = 64.87% Non-heterosexual = 35.13% Homosexual = 16.04% Other = 19.09% |
| Relationalstatus | In a relationship = 71.40% Single = 28.60% | In a relationship = 63.36% Single = 36.64% | In a relationship = 29.97% Single = 70.03% | In a relationship = 59.62% Single = 40.38% |

The current project was designed in accordance with the Declaration of Helsinki and received approval from the Ethical Committee for the Psychological Research of the University of Padova (Prot. n. 3049).

*2.2. Measures*

The online questionnaire, originally administered in Italian, consisted of 20 multiple-choice questions assessing demographic information, dating app use, and drug consumption (see Supplementary Materials for the questionnaire items).

Demographic information: The subjects were assessed for sex assigned at birth, age, educational level, relational status, and sexual orientation (it should be noted that we also asked for the participant's gender; however, we decided not to analyze this variable, as just 3.5% of the sample declared to be transgender, which made it impossible to run a sensible statistical analysis to compare the cisgender and transgender groups).

Drug consumption: The subjects were assessed for cannabis use (they were asked if they had used cannabis or marijuana during the past 12 months. If yes, they were asked how much: rarely (up to twice during the past 12 months); occasionally (up to once a month); frequently (up to once a week); daily. For the consumption of other illicit drugs, they were asked if they had used hard drugs during the past 12 months. If yes, they were asked how much: rarely (up to twice during the past 12 months); occasionally (up to once a month); frequently (up to once a week); daily.

Use of dating apps: The subjects were asked whether they were using (active users), had used but were no longer using (former users), or had never used any dating apps (non-users). The participants were informed that "dating apps" were intended as "online smartphone dating applications based on geosocial networking." If they were former users, they were asked for the duration of past use. Meanwhile, active users were asked for the age of the beginning of use and the primary motive for installing dating apps ("meet new people", "casual sex", "relationship", "transgression", "don't know"). The response options were in line with those from the study by Fowler and Both (Fowler and Both 2020). Transgression refers to the violation or contravention of implicit or explicit relational rules (e.g. extra-pair copulation in monogamous couples) or societal rules (e.g., writing or doing something that breaks social rules) (*Encyclopedia of Critical Psychology* 2014).

Intensity of dating app use: Active users were also assessed for the risk of the problematic use of dating apps (Bonilla-Zorita et al. 2020). They were asked about the number of times they accessed the apps, the frequency of notification checks, the frequency of stopping other activities to check the dating apps, the daily time spent on the dating apps, the perception of the uncontrolled use of the dating apps, the frequency of gaining access to

the apps without realizing this, the frequency of giving up hours of sleep to check the app notifications, the desire to reduce the amount of time spent on the apps, and the anxiety they felt if they were unable to use the apps.

A brief description of each analyzed variable is reported in the supplementary materials. In addition, data are available at the following repository: http://doi.org/10.5281/zenodo.5019220 (accessed on 20 July 2021).

### *2.3. Data Analysis*

To perform our data analysis, we used the open-source statistics programs JASP and R (JASP Team 2020; R Core Team 2020). Chi-squared ($\chi^2$) was run to test the association between categorical variables. We reported the odds ratio as the strength measure when the results were statistically significant. Note that for categorical variables with more than two levels of standardized residuals, ($z$) is reported instead of the odds ratio. If the value of $z$ lies outside of $\pm1.96$, then it is significant at $p < 0.05$; if it lies outside of $\pm2.58$, then it is significant at $p < 0.01$; if it lies outside of $\pm3.29$, then it is significant at $p < 0.001$ (Field et al. 2012). Pearson's correlation was performed to test the association between two continuous variables ($r$) or between a continuous and a dichotomized variable ($r_{pb}$).

For all of the statistical tests, the significance threshold of the $p$ value was set at 0.05, making explicit when the value was <0.01 and <0.001. However, for the purpose of addressing the problem of multiple testing, the Bonferroni correction was applied where needed, with the $p$-value divided by the number of tested variables.

Finally, multiple logistic regressions were run to investigate the contribution of multiple independent variables in the prediction of a categorical dependent variable. The collinearity assumption was checked before the model was run. The analysis was performed using the stepwise variable selection method.

## 3. Results

About a third (33.33%) of the participants admitted to using cannabis. However, just a few (6.57%) reported using it frequently or daily (the rate of subjects using soft drugs daily was 3.37%). A small number of participants admitted to using other drugs (3.68%). Just 0.39% (five subjects) declared using other drugs frequently. As all of them were non-users or former users, we will not further analyze these data. Table 2 shows the percentage of participants for each group (non-users, former users, active users) reporting using soft and hard drugs. This section is divided by subheadings. It provides a concise and precise description of the experimental results, their interpretation, and the experimental conclusions that can be drawn.

**Table 2.** Percentage of participants who declared using illicit drugs.

| | Non-Users | Former Users | Active Users | Overall Sample |
|---|---|---|---|---|
| Using cannabis | 27.59% | 38.93% | 37.63% | 33.33% |
| Using cannabis regularly [1] | 6.36% | 6.62% | 6.97% | 6.57% |
| Using other drugs | 3.34% | 4.58% | 3.14% | 3.68% |
| Using other drugs Regularly [1] | 0.50% | 0.51% | 0.00% | 0.39% |

[1] Regularly means up to once a week or daily.

### *3.1. Cannabis Use*

First, we investigated the role of demographic variables in cannabis consumption (Hypothesis 2). Note that the Bonferroni correction was applied; we divided the $p$-value by the number of tested variables (=5) and set the significance level at 0.01. No associations were found between sex assigned at birth and using cannabis ($\chi^2 = 0.11$, $p = 0.742$), or between sex assigned at birth and using cannabis regularly (frequently or daily) ($\chi^2 = 0.35$,

$p = 0.557$). Similarly, no statistically significant associations emerged between sexual orientation and using cannabis ($\chi^2 = 3.17$, $p = 0.075$), or between sexual orientation and using cannabis regularly (frequently or daily) ($\chi^2 = 0.12$, $p = 0.725$). When it comes to the role of age, the results showed a negative correlation between age and cannabis use ($r_{pb} = -0.10$, $p < 0.001$) but no correlation between age and using cannabis regularly ($r_{pb} = -0.02$, $p = 0.515$). Considering the education level, the results did not show any significant correlation between the years of school and using cannabis ($r_{pb} = -0.04$, $p = 0.121$) or using cannabis regularly ($r_{pb} = -0.01$, $p = 0.679$). However, a statistically significant association emerged between relational status (single vs. being involved in a relationship) and cannabis use ($\chi^2 = 14.06$, $p < 0.001$; odds ratio = 1.57). However, no associations emerged between relational status and using cannabis regularly ($\chi^2 = 0.50$, $p = 0.478$).

Second, we investigated the role of dating app use in cannabis consumption (Hypothesis 1). The results showed a statistically significant difference among non-users, active users and former users ($\chi^2 = 16.80$, $p < 0.001$). More precisely, the standardized residuals indicated that among people who had never used dating apps, significantly fewer cannabis users than expected were found ($z = -2.432$). In a comparison of app non-users and the other subjects (active users and former users), the odds of people using cannabis were 1.63 times higher between the active users and former users than the non-users ($\chi^2 = 16.67$, $p < 0.001$). No differences emerged between active app users and former app users ($\chi^2 = 0.12$, $p = 0.731$). However, these results were not consistent considering only the people who had declared using cannabis regularly (frequently or daily). Indeed, no statistically significant associations were found between using cannabis regularly and dating app usage (non-users vs. former users vs. active users) ($\chi^2 = 0.12$, $p = 0.941$).

As it emerged that cannabis consumption was associated with age, relational status and dating app use, a logistic regression analysis was run to investigate the role of each predictor (age, relational status, dating app use). The results are reported in Table 3.

**Table 3.** Output of logistic regression model featuring age, relational status, and dating app use (non-user vs. former user vs. active user) as predictors for cannabis use.

| | **B** | **SE** | **OR** | **Wald Test** | | | **95% Confidence Interval** | |
| --- | --- | --- | --- | --- | --- | --- | --- | --- |
| | | | | **Wald** | **df** | **p** | **Lower Bound** | **Upper Bound** |
| (Intercept) | 0.360 | 0.273 | 1.433 | 1.742 | 1 | 0.187 | −0.175 | 0.895 |
| Dating app use (non-user) | −0.546 | 0.141 | 0.579 | 15.088 | 1 | $1.026 \times 10^{-4}$ | −0.822 | −0.271 |
| Dating app use (active user) | −0.026 | 0.172 | 0.974 | 0.023 | 1 | 0.878 | −0.364 | 0.311 |
| Age | −0.034 | 0.009 | 0.967 | 14.724 | 1 | $1.245 \times 10^{-4}$ | −0.051 | −0.016 |
| Relational status (being single) | 0.293 | 0.131 | 1.340 | 4.997 | 1 | 0.025 | 0.036 | 0.549 |

Note. $R^2 = 0.03$ (McFadden), 0.03 (Nagelkerke), 0.003 (Tjur), 0.03 (Cox & Snell). Model Deviance = 1584.853, AIC = 1594.853, BIC = 1620.618, df = 1273, $\Delta\chi^2 = 4.981$, $p = 0.026$, AUC = 0.61. Step 1 (dating apps use): df = 1275, $\Delta\chi^2 = 16.905$, $p = 2.134 \times 10^{-4}$, $R^2 = 0.010$ (McFadden), 0.013 (Nagelkerke), 0.056 (Tjur), 0.013 (Cox & Snell); step 2 (dating apps use + age): df = 1274, $\Delta\chi^2 = 20.192$, $p = 7.004 \times 10^{-6}$, $R^2 = 0.023$ (McFadden), 0.029 (Nagelkerke), 0.025 (Tjur), 0.029 (Cox & Snell).

Regression analysis indicates that dating app use, age (being younger) and relational status (being single) were predictors of cannabis use and accounted for a significant proportion of the variance.

Then, we analyzed just the data of the active users, investigating the relationship between the app use pattern (the intensity of the use and length of dating app use; Hypothesis 3) and cannabis consumption. The Bonferroni correction was applied, with the *p*-value divided by the number of tested variables (=11), and with the significance level set at 0.005.

No statistically significant associations were found between cannabis use and the age of beginning to use apps (cannabis use in general [$r_{pb} = -0.09$, $p = 0.131$]; regular cannabis use [$r_{pb} = -0.05$, $p = 0.439$]). A negative trend was found between the number of years of app use and cannabis use in general [$r_{pb} = -0.14$, $p < 0.05$]; however, this trend did not reach statistical significance ($p < 0.005$). Similarly, no statistically significant associations

were found between the number of years of app use and regular cannabis use [$r_{pb} = 0.02$, $p = 0.789$].

Consistent trends of negative associations were found between users' responses regarding the intensity of using dating apps and the two variables of using cannabis and using cannabis regularly. More interesting, a statistically significant negative correlation was found between the general use of cannabis and the number of times of accessing the apps. Detailed results are reported in Table 4.

**Table 4.** Point-biserial correlations ($r_{pb}$) testing the association between the cannabis use variables (using cannabis and cannabis regularly) and the variables related to dating app use in the sample of active users. Note that when the Bonferroni correction is applied, the significance level is set at 0.005.

| Dating App Use | Using Cannabis | | Using Cannabis Regularly [1] | |
|---|---|---|---|---|
| | $r_{pb}$ | $p$ | $r_{pb}$ | $p$ |
| Amount of time per day spent on apps | −0.16 | <0.05 | −0.07 | 0.273 |
| Number of times of accessing the apps | −0.23 | <0.001 | −0.17 | <0.01 |
| Stopping other activities to check the apps | −0.18 | <0.01 | −0.12 | <0.05 |
| Frequency of checking notifications | −0.19 | <0.01 | −0.13 | <0.05 |
| Using dating apps more than desired | −0.17 | <0.01 | −0.07 | 0.220 |
| Accessing the app without realizing it | −0.06 | 0.330 | −0.06 | 0.313 |
| Giving up hours of sleep to check app notifications | −0.13 | <0.05 | −0.05 | 0.372 |
| Planning to reduce amount of time spent on the apps | −0.14 | <0.05 | −0.11 | 0.072 |
| Getting anxious or missing something if using the apps is not possible | −0.18 | <0.01 | −0.13 | <0.05 |

[1] Regularly means up to once a week or daily.

In analyzing the motives for app installation, we applied the Bonferroni correction, dividing the *p*-value by the number of tested variables (=5) and setting the significance level at 0.01. We did not find any statistically significant correlation with the general or regular use of cannabis. However, it is worth noting a trend showing that the number of active users who consumed cannabis was smaller among those who had installed apps to find friends (odds ratio = 0.37). Moreover, among the active users who regularly consumed cannabis, more people installed dating apps for transgression purposes (odds ratio = 3.89), and fewer people installed them to find romantic partners (odds ratio = 0.16). The results are reported in Table 5.

**Table 5.** Results from chi-squared analysis investigating the association between different levels of using cannabis and app installation motives in the sample of active users. Note that when the Bonferroni correction is applied, the significance level is set at 0.01.

| Motives for Installing Dating Apps | Using Cannabis | | | Using Cannabis Regularly [1] | | |
|---|---|---|---|---|---|---|
| | $\chi^2$ | $df$ | $p$ | $\chi^2$ | $df$ | $p$ |
| Meeting new people | 4.124 | 1 | <0.05 | 2.142 | 1 | 0.143 |
| Beginning a relationship | 2.254 | 1 | 0.133 | 4.043 | 1 | <0.05 |
| Having casual sex | 1.097 | 1 | 0.295 | 0.014 | 1 | 0.905 |
| Transgression | 1.401 | 1 | 0.236 | 5.631 | 1 | <0.05 |
| I don't know | 0.922 | 1 | 0.337 | 2.287 | 1 | 0.130 |

[1] Regularly means up to once a week or daily. Note: odds ratio = 0.37 (using cannabis and app installation to meet new people), 0.16 (using cannabis regularly and app installation to begin a relationship), 3.89 (using soft drugs regularly and app installation for transgression purposes).

Finally, among the former users, no associations were found between the past use of dating apps (more than six months) and cannabis use ($\chi^2 = 0.09$, $p = 0.768$), and between

being a former dating app user and the regular consumption of cannabis ($\chi^2 = 1.90$, $p = 0.168$).

### 3.2. Other Drugs

First, we investigated the role of demographic variables in hard drug consumption (Hypothesis 2). Note that the Bonferroni correction was applied, with the *p*-value divided by the number of tested variables (=5) and setting the significance level at 0.01. No statistically significant results emerged during the investigation of the association between sex assigned at birth and hard drug use ($\chi^2 = 0.36$, $p = 0.550$). With regard to sexual orientation, no associations were found between sexual orientation and hard drug use ($\chi^2 = 0.21$, $p = 0.643$). Moreover, the results did not highlight any statistically significant correlation between hard drug use and age ($r_{pb} = 0.01$, $p = 0.641$). Similarly, no significant correlations emerged between educational level and using hard drugs ($r_{pb} = -0.01$, $p = 0.764$). No associations emerged between relational status (single vs. involved in a relationship) and hard drug use ($\chi^2 = 1.49$, $p = 0.223$).

Second, we investigated the role of dating app use in hard drug consumption (Hypothesis 1). No associations were found between dating app use (non-users vs. active users vs. former users) and hard drug use ($\chi^2 = 1.33$, $p = 0.515$). Indeed, active users were likely to consume hard drugs just as others (former users and non-users) were ($\chi^2 = 0.31$, $p = 0.580$).

Next, we analyzed just the data on the active users, investigating the relationship between the app use pattern (the intensity of the use and length of dating app use; Hypothesis 3) and hard drug consumption. The Bonferroni correction was applied, with the *p*-value divided by the number of tested variables (=11) and the significance level set at 0.005. No statistically significant correlations were found between hard drug use and the age of beginning to use the apps ($r_{pb} = 0.01$, $p = 0.893$), or the number of years of dating app use ($r_{pb} = -0.03$, $p = 0.643$). In addition, no significant correlations were found between hard drug use and all of the variables, thus indicating the intensity of using dating apps among active users (see Table 6).

**Table 6.** Point-biserial correlations ($r_{pb}$) testing the association between hard drug use and the variables related to dating app use in the sample of active users.

| Dating Apps Use | Using Other Drugs | |
|---|---|---|
| | $r_{pb}$ | $p$ |
| Amount of time per day spent on apps | −0.09 | 0.143 |
| Number of times apps were accessed | −0.05 | 0.358 |
| Stopping other activities to check apps | −0.04 | 0.466 |
| Frequency of checking notifications | −0.08 | 0.158 |
| Using dating apps more than desired | −0.03 | 0.627 |
| Accessing the app without realizing it | −0.02 | 0.788 |
| Giving up hours of sleep to check app notifications | −0.03 | 0.606 |
| Planning to reduce amount of time spent on the apps | −0.01 | 0.878 |
| Getting anxious or missing something if using the apps is not possible | −0.003 | 0.961 |

In analyzing the motives for app installation, we applied the Bonferroni correction, dividing the *p*-value by the number of tested variables (=5) and setting the significance level at 0.01. We did not find any statistically significant association between the motivations for installing dating apps and hard drug use (meeting new people: $\chi^2 = 0.93$, $p = 0.336$; beginning a relationship: $\chi^2 = 2.83$, $p = 0.093$; casual sex: $\chi^2 = 0.25$, $p = 0.617$; I don't know: $\chi^2 = 0.01$, $p = 0.941$) except for in the case of active users who declared having installed the applications for transgression purposes ($\chi^2 = 9.96$, $p < 0.01$; odds ratio = 7.56), as the odds of taking hard drugs between them were 7.56 times higher than that for users who had installed them for other reasons.

In analyzing the data from former users, we did not find a relationship between hard drug consumption and the use of dating apps for a long period of time (more than six months) ($\chi^2 = 0.22$, $p = 0.642$).

## 4. Discussion and Conclusions

The current study explored, for the first time, the association between the use of mobile dating apps and drug consumption (cannabis versus other drugs), not in conjunction with sexual activities, in the general population. The study investigated differences among app users, non-users and former users, considering the moderating role of demographics, as well as patterns of dating app use (the intensity of use and motives).

With regard to cannabis use, the results from the present study indicated an association between dating app use and cannabis consumption. The study findings are in line with previous studies (Erevik et al. 2020; Fansher and Eckinger 2020; Phillips et al. 2014). Conversely, the consumption of drugs different from cannabis was not associated with dating app use. Therefore, Hypothesis 1 was partially confirmed. However, when we focused on the pattern of drug consumption, although cannabis use in general was associated with dating app use, regular cannabis consumption was not. These findings contribute to enriching the current knowledge about the relationship between dating app use and drug consumption, and they suggest the need to differentiate between the consumption of cannabis and other drugs, and between occasional and regular cannabis users. Moreover, they are in line with previous studies on drug consumers' psychosocial characteristics and personality features (Tang et al. 1996; Brook et al. 2016; Patton et al. 2002; Cascone et al. 2011; Rey et al. 2002; Copeland and Swift 2009). In alignment with the literature, occasional cannabis use may be associated with experimentation, curiosity, and seeking novelty, whereas regular use, as well as "hard drug" consumption, may be associated with emotional dysregulation and significant psychological distress (i.e. mood disorders or personality disorders, etc.) (Kedzior and Laeber 2014; Weinberger et al. 2019; Karila et al. 2014; Rosenberg 2019; Lee et al. 2018; Brook et al. 2016). As dating apps are tools used to meet new people, and as previous studies highlighted that dating app users tend to be more extraverted and open to new experiences compared with non-users (Timmermans and De Caluwé 2017), we may assume that seeking novelty is a possible common underlying factor between dating app use and cannabis use. This might also be consistent with what emerged about age in this study: young people were more prone to consuming cannabis occasionally than adults were, but they were not more prone to using it regularly. Indeed, the literature highlighted that adolescents and young adults are more prone to seeking novelty and experimentation (Wolfe et al. 2006).

When it comes to the influence of socio-demographic variables (Hypothesis 2), being younger and being single were associated with a higher odds of cannabis consumption. The results regarding the role of age confirm our Hypothesis 2. Conversely, many interpretations are possible for the association between being single and cannabis use. Probably, among cannabis users, some latent psycho-relational features are commonly linked to both being single and using cannabis. In this regard, the literature indicates that people with higher levels of loneliness (Rokach and Orzeck 2003), lower levels of self-esteem (Olmstead et al. 1991), and anxious attachment orientations (Fairbairn et al. 2018) are more likely to use substances of abuse. In contrast with our Hypothesis 2, being male was not associated with a higher odds of drug consumption (both cannabis and other drugs). Concerning this point, a recent gender-based review of addictive disorders highlighted that the differences in the prevalence rates between genders are getting narrower (Fonseca et al. 2021).

On the contrary, no significant associations emerged between the consumption of other drugs (drugs other than cannabis) and any socio-demographic variables. This suggests that hard drug use might primarily depend on personality features or on personal life-event variables, quite independently from socio-demographics. Therefore, our Hypothesis 2 was partially confirmed. These results also substantiate the value of differentiating between cannabis use and the use of other drugs.

The results regarding the intensity of app use showed a particular trend. Indeed, beyond the general association between dating app use and cannabis consumption, the heavy app users (those using the apps in an addictive-like way), particularly those with greater access to the apps, tended to be less likely to use cannabis compared with other dating app users. These data are in contrast with other studies' data on the relationship between problematic internet use or smartphone addiction and substance use (Bakken et al. 2009; Padilla-Walker et al. 2010; Lee et al. 2013). However, the data are consistent with previous studies regarding the association between dating app use and smoking. Although smokers were more likely to be active app users, people using the apps intensively were less likely to smoke (Flesia et al. 2021a). These results might depend on the significant psychological impact that dating apps can have on personal experience. Unlike the internet or other social media, dating apps are preferably accessed through the smartphone and are GPS based. Therefore, they can be used anytime and everywhere, constantly providing users with the ability to connect, and to feel connected, with plenty of potential partners located near them. Through these features, dating apps can assume a significant psycho-relational value for their users. According to the "Uses and Gratifications" theory (Katz et al. 1973; Whiting and Williams 2013), media use satisfies users' social and psychological needs. According to the "Media Practice Model" (Steele and Brown 1995), media use is a function of the dialectical interaction between media characteristics and the user's individual characteristics. In this regard, consistent with the "recreation hypothesis", people scoring high in sensation seeking were more prone to using dating apps (Peter and Valkenburg 2007). Sensation seeking is "a trait defined by the seeking of varied, novel, complex, and intense sensations and experiences and the willingness to take physical, social, legal, and financial risks for the sake of such experiences" (Zuckerman 1994). Evidence exists that sensation seekers use dating apps to meet new people, have casual sex, and enjoy hookups (Peter and Valkenburg 2007; Chan 2017). Moreover, the literature reports an association between sensation seeking and addictive behaviors (Tapia León et al. 2019). People scoring high in social anxiety are also more likely to use dating apps. According to the "compensation hypothesis" (Peter and Valkenburg 2007), people high in social anxiety seek dates online or spend time online because the features of online communication (e.g. reduced cues, anonymity, controllability) allow them to compensate for the deficits they experience in offline dating. Cannabis use is also associated with both sensation seeking and social anxiety (Rahm-Knigge et al. 2019). Therefore, among dating app users who intensely use the apps and are characterized by specific personality features (i.e., sensation seeking, social anxiety), dating app use might become a substituting activity, replacing or buffering the type of effect that is otherwise linked to cannabis use. Future efforts are needed to investigate these issues. The results regarding the intensity of dating app use confirm our Hypothesis 3. The length of use, conversely, was not associated with differences in the odds of drug consumption. This result is in contrast with previous studies on the association between the length of dating app use and the odds of engaging in sexual activity involving recreational drugs (Rogge et al. 2020). Thus, Hypothesis 3 is partially not confirmed. However, the result confirms the value of investigating the association between dating app use and drug consumption even outside of sexual activities.

Although the results regarding the motives for installing the apps did not show any significant association with drug consumption, a trend in the results suggests the partial role of motives in influencing the association between dating app use and drug use, distinguishing between dating app users (Timmermans and De Caluwé 2017; Sumter et al. 2017). More specifically, people installing the apps with the primary aim of "meet new people" or "begin a relationship" tended to be less prone to using cannabis, suggesting that people primarily consider the apps to be tools for finding friends or relationships. This is different from people who use them with other motives (i.e., to find casual sex or for transgression purposes). Differences in personality-based antecedents may play a common underlying mediating role. Consistently, the transgression motive tended to be associated with higher odds of both regular cannabis use and "hard drugs" consumption. As no

specific personality scales or mood scales were administered, further research is needed to test these possible explanations. In this sense, the present exploratory study can serve as a guide for the planning of future related lines of research.

Our findings give interesting cues for possible preventive campaigns targeting dating app users not engaged in relationships. For instance, dating apps' registration or login pages could ask users their relational statuses; then, dating apps could add links to information regarding the factors associated with drug consumption, especially for single users.

The current study has some limitations for consideration. The participants were recruited through an online link posted and advertised on social media. Online advertising guarantees the possibility of recruiting large samples, but it may not guarantee the sample representativeness. In addition, the online self-reported format guarantees anonymity, but it does not allow for verifying the reliability of responses and the understanding of questions from participants. Moreover, data regarding "hard drug" users come from a small sample, thus limiting the generalizability of results. Finally, we investigated participants' drug consumption during the past 12 months. Distinguishing among active, former and non-use when it comes to drug consumption and dating app use could provide further information regarding the association between dating app use and drug consumption.

To conclude, the present study contributes to enriching the limited literature in this research field, highlighting some associations between dating app use and drug consumption, and suggesting possible pathways of explanations. The study suggests a possible substitutive effect of dating apps in people who heavily use the apps. Future efforts are needed to identify possible explanatory factors. Further efforts are also needed to better understand the psychological mechanisms underlying these associations. In this sense, further research could investigate the influence of common personality-based antecedents (i.e., impulsivity, novelty and sensation seeking, attachment orientations) and their interactions with motives for using the apps and consuming drugs.

**Supplementary Materials:** The following are available online at https://www.mdpi.com/article/10.3390/socsci10080290/s1, Questionnaire items; List of variables.

**Author Contributions:** Conceptualization, L.F. and M.M.; methodology, L.F. and M.M.; software, M.M. and V.F.; validation, L.F. and M.M.; formal analysis, M.M. and V.F.; investigation, L.F.; resources, L.F. and M.M.; data curation, L.F., V.F. and M.M.; writing—original draft preparation, L.F., V.F. and M.M.; writing—review and editing, L.F., V.F., C.F. and M.M.; visualization, L.F., V.F., C.F. and M.M.; supervision, L.F.; project administration, L.F. and M.M.; funding acquisition, L.F. and C.F. All authors have read and agreed to the published version of the manuscript.

**Funding:** This research received no external funding.

**Institutional Review Board Statement:** The current project was designed in accordance with the Declaration of Helsinki and received approval from the Ethical Committee for the Psychological Research of the University of Padova (Prot. n. 3049).

**Informed Consent Statement:** Please add "Informed consent was obtained from all subjects involved in the study."

**Data Availability Statement:** Data available at the following repository: http://doi.org/10.5281/zenodo.5019220 (accessed on 20 July 2021).

**Conflicts of Interest:** The authors declare no conflict of interest.

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
