# Peer review of "The Relationship between Drug Consumption and Dating App Use: Results from an Italian Survey"

_socsci, doi:10.3390/socsci10080290_

Round 1

Reviewer 1 Report

I found the article very interesting both for the topic and the results. The literature review seems to be up-to-date and comprehensive. The research hypotheses are clearly presented.
The section on methodology adequately describes the procedures and the sample.
The results are clearly presented and the statistical analysis works adequately. 
The section I would work on is the discussion section where the authors introduce issues such as "personality traits" by presenting them as variables that could at least partly explain the results. The reader wonders why a personality scale (BIG-FIVE, for example) was not administered. Between the limitations and future developments of the research, this issue needs to be better argued. 

Similarly to the relationship between "extroversion" and the use of dating apps, the relationship between "being single" and "loneliness", "lower level of self-esteem", "anxious attachment", is not argued. Also in this case, as for the previous one, the questionnaire administered does not allow to make this association.

The first question that arises is: why did the authors not use a scale to measure these traits? For example, why did the authors not use the IPPA to assess attachment parenting? Or even why did they not use a scale to assess anxiety?  They did not use these scales and do not mention this lack among the limitations.

Minor comments

In the word file in which the authors present the items of the questionnaire, they refer to a 'table', which however does not appear.

On line 356 there is a sentence in which the words “on” and “of” appear. This sounds wrong.

On line 436 there is something syntactically wrong with the sentence. There is one point (dot) that probably needs to be deleted.

Author Response

Dear Reviewer, we thank you for your careful reading of the manuscript and helpful comments and suggestions. We have made revisions and highlight the changes in the manuscript according to your comments and suggestions, as described below.

I found the article very interesting both for the topic and the results. The literature review seems to be up-to-date and comprehensive. The research hypotheses are clearly presented.
The section on methodology adequately describes the procedures and the sample.
The results are clearly presented and the statistical analysis works adequately. 
The section I would work on is the discussion section where the authors introduce issues such as "personality traits" by presenting them as variables that could at least partly explain the results. The reader wonders why a personality scale (BIG-FIVE, for example) was not administered. Between the limitations and future developments of the research, this issue needs to be better argued. Similarly to the relationship between "extroversion" and the use of dating apps, the relationship between "being single" and "loneliness", "lower level of self-esteem", "anxious attachment", is not argued. Also in this case, as for the previous one, the questionnaire administered does not allow to make this association. The first question that arises is: why did the authors not use a scale to measure these traits? For example, why did the authors not use the IPPA to assess attachment parenting? Or even why did they not use a scale to assess anxiety?  They did not use these scales and do not mention this lack among the limitations.

Response: Thank you for arising this issue. We did not administer further personality scales, because in this exploratory study we wanted to investigate the possible relation between dating app use and drugs use. Certainly, results from the study pave the way for further investigating these issues. We added this consideration in the Discussion section.

Minor comments

In the word file in which the authors present the items of the questionnaire, they refer to a 'table', which however does not appear.

Response: Thank you for this remark. We changed “table” with “list”.

On line 356 there is a sentence in which the words “on” and “of” appear. This sounds wrong.

Response: Thank you for this remark. It has been amended.

On line 436 there is something syntactically wrong with the sentence. There is one point (dot) that probably needs to be deleted.

Response: Thank you. It has been amended.

Reviewer 2 Report

It is a very interesting paper, in terms of the theme, methodology, analysis and reflection. Overall, it is very good.

Comments the author could take in account:

  • It would be interesting to know previous data about app users such as percentage of users in general population or profiles;
  • Also, using three profiles of apps user, we could consider drug users as active, former and non-users;
  • It is hard to understand a linear association between dating apps and drug use: there are other motives for installing dating apps rather then to use drugs; generally drug use is either taken alone or along with friends and not so much with strangers; it is rather difficult to define drug use as the dependent variable as using dating apps is also a behavior (why the odds of drug use associated with the dating apps use and not the other way around?).

Congratulations!   

Author Response

Dear Reviewer, 

we thank you for your careful reading of the manuscript and helpful comments and suggestions. We have made revisions and highlight the changes in the manuscript according to your comments and suggestions, as described below.

It is a very interesting paper, in terms of the theme, methodology, analysis and reflection. Overall, it is very good.

Response: Thank you for your acknowledgment.

Comments the author could take in account:

  • It would be interesting to know previous data about app users such as percentage of users in general population or profiles;

Response: Thank you for this suggestion. We added this sentence in the Introduction section: “The number of dating app users has been increasingly growing in recent years: according to the Statista Digital Market Outlook, there were 28.9 million users of online dating services in the United States in 2017, they raised to 44.2 million in 2020, and are estimated to reach 53.3 million by 2025. Globally, 270 million people worldwide used dating apps in 2020. Tinder, the most popular dating app, counts 1.6 billion “swipes” every day.

  • Also, using three profiles of apps user, we could consider drug users as active, former and non-users;

Response: Thank you for this interesting consideration. Certainly, distinguishing drug consumption between active, former and non-use could give further information in regard of the association between dating app use and drugs consumption. We added this suggestion in the Discussion section.

  • It is hard to understand a linear association between dating apps and drug use: there are other motives for installing dating apps rather then to use drugs; generally drug use is either taken alone or along with friends and not so much with strangers; it is rather difficult to define drug use as the dependent variable as using dating apps is also a behavior (why the odds of drug use associated with the dating apps use and not the other way around?).

Response: We agree with the reviewer’s observation; indeed, in the paper we always about association, rather than causal relationships between dating app use and drug consumption. Investigating potential common motives, underlying both dating app use and drug use could be a very interesting topic. In this sense, results from the present paper opened a new line of research. We added this consideration in the Discussion section.